# Motor Synergies Measurement Reveals the Relevant Role of Variability in Reward-Based Learning

**DOI:** 10.3390/s21196448

**Published:** 2021-09-27

**Authors:** Carla Caballero, Francisco J. Moreno, David Barbado

**Affiliations:** Sport Sciences Department, Miguel Hernández University of Elche, 03202 Elche, Spain; ccaballero@umh.es (C.C.); dbarbado@umh.es (D.B.)

**Keywords:** computer simulated task, learning ratio, throwing task, dynamometer, force sensor

## Abstract

Currently, it is not fully understood how motor variability is regulated to ease of motor learning processes during reward-based tasks. This study aimed to assess the potential relationship between different dimensions of motor variability (i.e., the motor variability structure and the motor synergies variability) and the learning rate in a reward-based task developed using a two-axis force sensor in a computer environment. Forty-four participants performed a pretest, a training period, a posttest, and three retests. They had to release a virtual ball to hit a target using a vertical handle attached to a dynamometer in a computer-simulated reward-based task. The participants’ throwing performance, learning ratio, force applied, variability structure (detrended fluctuation analysis, DFA), and motor synergy variability (good and bad variability ratio, GV/BV) were calculated. Participants with higher initial GV/BV displayed greater performance improvements than those with lower GV/BV. DFA did not show any relationship with the learning ratio. These results suggest that exploring a broader range of successful motor synergy combinations to achieve the task goal can facilitate further learning during reward-based tasks. The evolution of the motor variability synergies as an index of the individuals’ learning stages seems to be supported by our study.

## 1. Introduction

Motor variability plays a functional role in human adaptive behaviors, related to the facilitation of motor learning [1,2]. Several studies support the idea that the Central Nervous System (CNS) modulates motor variability to enable the exploration of all the possible configurations provided by the large number of motor system degrees of freedom (DOF), enhancing the achievement of the required movement solution [3,4,5]. How the CNS regulates motor variability seems to depend on both individual and environmental characteristics [6,7]. On the one hand, some studies have supported the idea that different levels of variability should be manipulated according to the individual’s learning stage. It seems that there is a need for inducing higher-variability conditions to promote motor variations when exploration is required to learn a novel task. In contrast, lower-variability conditions would facilitate a more consistent motor output which, in turn, would improve motor performance when exploiting a viable solution [2,8]. On the other hand, there are some differences in the effect of variability in the learning process according to the type of task. For instance, reward-based learning is based on the idea that if an action is followed by a successful output, the tendency to repeat that same action is strengthened [9]. However, in this learning condition, the output received by the individual who performs the action only indicates how successful that output was; and, thus, it carries no other feedback information about the motor execution that allows the individual to modify their motor behavior [10]. This reason explains why this type of learning is characterized by the need for exploration to gather knowledge about the optimal solution and exploitation of the knowledge accrued to keep performing the right solution once it is found [2]. Thus, higher motor variability has been related to faster reward-based learning [2,11].

On the other hand, error-based learning consists of modifying the motor behavior based on the perception of the difference between the planned and the observed motor output. Thus, the improvement during practice is based on the individual’s attempt to cancel or, at least, reduce the perturbation through the production of an output that counteracts the current estimated output [12]. In this type of learning, the potential benefits promoted by higher levels of motor variability are not as clear as in reward-based learning [2].

In this study, we focused on the role of motor variability in a reward-based learning task to estimate whether motor variability is a consequence of noise (which must be reduced in order to improve motor performance), or whether it plays a functional role to ease learning.

To assess the reward-based learning processes properly it is necessary to design tasks that reduce the potential influence of error-based learning mechanisms. Laboratory tasks designed following the reward-based learning paradigm are commonly based on computer simulation tasks in which “shooting” or “reaching” are required to aim at targets that are not visible by means of different sensors (e.g., dynamometers, inertial sensors, etc.) [13]. This experimental setup enables not only analyzing the probability of success during the reward-based task but also assessing the variability of the motor behavior that leads to a successful or unsuccessful performance. Among the mathematical methods to assess motor variability, nonlinear tools measure the temporal dynamics (i.e., the structure of variability) of those signals obtained from the sensors. Nonlinear tools refer to all those mathematical measures that quantify the temporal organization of the motor dynamic through the degree to which the values of a signal time series emerge or change in an orderly manner, often across a range of time scales [14]. The structure of motor variability has been previously shown as a relevant index to identify how an individual explores the environment to promote learning [15,16]. Detrended fluctuation analysis (DFA) examines long-range autocorrelation assessing the extent to which further motor behavior is dependent on previous fluctuations [17,18]. Lower long-range autocorrelation of movement fluctuation has been related to higher flexibility to perform motion adjustments [17,19], and it has been successfully used to predict motor learning rate in an error-based learning balance task [1]. However, to the best of the authors’ knowledge, it is unclear whether the analysis of variability structure can be also used to predict motor learning rate in reward-based learning.

Other mathematical methods have examined the movement variability of motor synergies to analyze the shape of the different repeated executions in the different dimensions of the solution space, such as the uncontrolled manifold (UCM), the goal-equivalent manifold or the tolerance, noise and covariation approaches [20]. These methods are based on the identification of the solution manifold, defined as the possible motor configurations that provide a successful performance and how the different repeated executions are distributed around it. Accordingly, these methods are capable of discriminating between “good” and “bad” variability (GV and BV, respectively) depending on whether the variations distributions are or not in the range of the redundant solutions of a motor problem. The ratio between these two types of variability, which provides information about motor synergies, has been related to different learning stages [21], strengthening of motor synergies, and it is characterized by the elaboration of an adequate referent configuration trajectory and the elaboration of multijoint (multimuscle) synergies. The second one consists in the weakening of those motor synergies when other aspects of motor performance are optimized. In turn, Latash et al. (2002) suggested that the relationship between GV and BV could quantify how the system controls the different DoFs, indicating whether the system displays a motor synergy or not or how strong this synergy is. Combining these two ideas, the first stage of motor learning described would be characterized by the finding and refining of the motor control needed in the task, emerging and strengthening motor synergies with practice, while the second stage would be related to a drop of the synergy strength because in the phase the aim is to optimize other aspects of performance.

To summarize, the practice would cause changes in the ratio between good and bad variability, which in turn, would be related to how the degrees of freedom are coordinated in the different learning stages and the prevalence of the exploration or exploitation of motor synergy strategies [22,23,24]. However, although the relationship between the motor synergies variability and learning stages has been previously addressed [21], to the best of our knowledge, no studies have analyzed how motor synergies variability conditions the learning rate, especially in reward-based motor tasks.

Therefore, the main aim of this study was to assess the relationship between motor variability and learning rate in a reward-based learning task. The application of a two-axis force sensor in a virtual computer task enabled measurement of the features of motor variability through its structure and the motor synergies characteristics. Based on the literature, the hypothesis of this study is that both the motor variability synergies and the motor variability structure will be related to reward-based learning processes. A higher ratio between “good” and “bad” variability, as well as a lower autocorrelated variability structure, will be related to a higher learning rate.

## 2. Materials and Methods

### 2.1. Participants

Forty-four healthy participants (11 females, 33 males) took part in this study (age = 26.46 ± 6.03 years; stature = 1.74 ± 0.08 m; mass = 69.90 ± 11.04 kg). Exclusion criteria included musculoskeletal injuries that impaired participants from pushing a vertical handle with their hands applying a minimum of force or visual deficits that impaired participants from seeing the visual information provided by a computer screen correctly (screen size = 21”, distance of the handle to the screen = 48 cm).

Written informed consent was obtained from each participant prior to testing. Data were treated anonymously, and all participants were informed of the risks and benefits of the trial.

### 2.2. Instruments and Procedure

Participants performed a computer simulation task in which they had to use a vertical fixed handle attached to a dynamometer (FSSB-R3 Warhog, RealSimulator, Madrid, Spain) to release a virtual ball to hit a target. FSSB-R3 is a two-axis force sensor (medial lateral and anterior posterior forces, 98 mm × 98 mm × 60 mm; 350 gr) with a maximum sensitivity of 0.025 lb and a maximum allowed force of 20 lb. The FSSB R3 includes an electronic calibration that allows the user to perform this operation as many times as necessary. The FSSB-R3 was connected by USB to the computer and the sample rate was set at 100 Hz.

The participants held the handle and pushed it when they wanted to release a virtual ball from the bottom of the screen towards the target. The target was located in the upper-left corner (45°) but was not visible for the participants (Figure 1). The way participants could regulate the direction of the ball to hit the target was combining the forces applied in the medial lateral (MLf) and the anterior posterior (APf) axes. Participants could not see the ball trajectory nor where the ball ended. They just saw a green light if they hit the target (Figure 1A), or red light if they did not (Figure 1B). The task was specifically designed by the researchers for this study using Labview Software v.11 (National Instruments, Austin, TX, USA).

The participants performed the task with their dominant hand, seated on a chair and with their arm and forearm aligned in front of the handle. The sitting position and distance of the participant to the screen was at the participant’s choice (Figure 2).

Each participant performed a pretest, a training period, a posttest, and three retests (10 min, 24 h, and 1 week apart). Each test consisted of 100 trials and the training period consisted of 6 series of 100 trials each.

In order to assess throwing performance for each trial, the ball trajectory was computed through the dynamometer FSSB-R3 connected to the handle. The combination of anterior–posterior and medial–lateral forces gave the release angle. An application written in Labview was used for data collection.

### 2.3. Data Analysis and Reduction

An application written in MatLab R2020a (MathWorks, MA, USA) was used to compute the dependent variables in this experiment. The force magnitude (FM) was computed as the result of the forces in both axes at the moment of the ball release. The hit ratio (HR) was used as the main parameter to assess participants’ accuracy. It was computed as the number of times the participants were able to throw the ball at the range angle to hit the target. To measure the effect of practice, the learning ratio (LR) was computed by the differences in the HR between the pretest and the tests performed after the practice period (posttest and the three retention tests). The time series of data from the relative error of each throw was also extracted, calculating the positive or negative difference between the release angle of the ball in each trial and the optimal angle that successfully would allow to hit the target.

To address the variability in the combination of the horizontal forces that resulted in the release angle of the virtual ball, an adaptation of the procedure applied by Gates and Dingwell (2008) [25], based on the goal equivalent manifold approach introduced by Cusumano and Cesari (2006) [26], was applied. This procedure assumes that there is an infinite number of combinations of forces in the mediolateral and anteroposterior axes that can result in the successful angle corresponding to the center of the target (i.e., 135°, 2.35619 rad) that allows the participant to achieve the goal (hitting the target). The space of these two elemental variables is two-dimensional (a plane), and the magnitude of their adequate combinations may be represented as a one-dimensional subspace (the solid line in Figure 3). These *MLf**–APf* combinations define the goal equivalent manifold (GEM) of the throwing task that corresponds to the successful angle [tan*^−^*^1^ (*MLf*/*APf*)] required to hit the target. As long as the system stays on that line, the task is successfully performed. The *MLf**–APf* successful combination was defined as the line defined by the following slope–intercept form equation:(1)$$APf=m\times MLf+APf_{0}$$
where *APf* and *MLf* are the forces in the anterior–posterior and medial–lateral directions, *APf*_0_ is the AP-intercept of the line with an arbitrary value of 0, and m is the line slope defined as the tangent of 2.35619 rad (*m* = 1.00). After this assumption, the variability can be decomposed into components or combinations of forces that can result in the effective angle to hit the target (good variability or GV) or not (bad variability or BV), resulting in a variability tangent to and perpendicular to the GEM, respectively. In this experiment, GV was computed as the standard deviation of the distance between the force combination in each trial and the GEM line. BV was computed as the standard deviation of the orthogonal distance between the force combination in each trial and the synergy line. The ratio between the good and bad Variability (GV/BV) was computed as the division between the GV by the BV. Higher values are interpreted as stronger synergy in variability (see Figure 3). Commonly, it has been interpreted that participants who display higher GV, have higher flexibility in their motor patterns, since they are able to achieve the task goal using different successful combinations. However, it has been interpreted that those participants who display higher BV are showing an exploratory behavior in which they are still looking for the right solutions. Thus, an increment in performance would be followed by an increment in the GV/BV [22].

Finally, the detrended fluctuation analysis (DFA) was computed from the relative error of the throwing series to assess the motor output variability structure. DFA represents a modification of the classic root mean square analysis with a random walk to evaluate the presence of long-term correlations within a time series using a parameter referred to as the scaling index α [18,27]. The scaling index α corresponds to a statistical dependence between fluctuations at one timescale and those fluctuations over multiple timescales. This procedure assesses the extent to which further motor behavior is dependent on previous fluctuations [28]. Less dependency on previous behavior (lower long-range autocorrelation; lower α) has been interpreted as a higher flexibility to perform motion adjustments [1]. This measure was computed according to the procedures of Peng et al. (1995). In this study, the slope α was obtained using a window range of N/10, going from 5 ≤ *n* ≤ 15 to maximize the long-range correlations and reduce errors incurred in by estimating α [29]. Different values of α indicate the following: α > 0.5 implies persistence in the position (the trajectory tends to remain in its current direction); α < 0.5 implies antipersistence in the position (the trajectory tends to return to where it came from) [27].

### 2.4. Statistical Analysis

Normality of the variables was evaluated using the Kolmogorov–Smirnov test with the Lilliefors correction. First of all, repeated-measures ANOVAs were used to assess the effect of reward training on all the variables measured. Pearson Product Moment Correlation coefficients were calculated to analyze the relationship between the initial performance, variability parameters, and the learning ratio. The correlational analysis revealed a relationship between the initial performance, initial variability level, and learning rate (see results section to check the correlational results). After that, it was decided to group the participants using a linear regression method (see Barbado et al. (2017) for more information) to assess if higher or lower initial variability was related to different learning rates, avoiding the potential bias caused by the initial performance. First, participants were divided into three groups according to their initial performance level, consisting of the lowest, intermediate, and highest HRPRE scores. Then, a linear regression was made between initial performance (i.e., HRPRE) and the representative variables for analyzing motor variability (i.e., GV/BV and DFA). Finally, participants were grouped according to their residual scores. Specifically, participants showing negative residuals scores (i.e., lower variability scores than the scores predicted by the regression analysis according to the participants’ initial performance) were included in the “lower variability group”. Conversely, participants showing positive residual scores (i.e., higher variability scores than the scores predicted by the regression analysis according to the participants’ initial performance) were included into the “higher variability group”. Therefore, the whole sample was split into two different groups with similar initial performance level but with different initial variability levels. A mixed-way ANOVA was performed with HR as the within-subject factor and initial variability as the between-subject factor. All statistical analyses were performed using IBM SPSS software 26, with a significance level set at *p* < 0.05.

## 3. Results

Average values obtained in the different evaluation tests and the repeated measures ANOVA results are displayed in Table 1 (for checking the whole database, please, see the Appendix A). Apart from the GV/BV and the DFA, all the dependent variables showed significant differences caused by the training.

The pair comparisons results (Figure 4) showed significant differences between the tests in the experimental variables. Regarding FM (Figure 4A), immediately after training, the values were reduced and maintained in the Retest1 but they increased back to their previous values in the Retest2 and Retest3. Performance increased (i.e., higher HR) after training and these higher values were maintained even in the Retest3 (Figure 4B). Regarding the variability measures, DFA showed slightly higher mean values after practice and it decreased in the subsequent retention tests, but no significant differences were observed (Figure 4C). The GV and BV decreased after training, but the GV/BV remained stable along the measurements (Figure 4D–F). In the Retest2, GV started to increase, but it did not show significant differences with the previous evaluations. BV values were still significantly lower than in the pretest. Finally, in the Retest3 both GV and BV moved closer to the pretest values.

Correlational analyses (Table 2) showed a negative correlation between the initial performance (HRPRE) and the GV and BV initial levels (GVPRE, BVPRE) with moderate and strong correlation indexes, respectively. In addition, the GV/BV in the pretest (GV/BVPRE) was also related to the initial performance in a positive way, showing a strong correlation. Regarding the relationship with the learning ratios, those participants who displayed lower HRPRE were those who learned more. Regarding the variability measures, BVPRE values correlated directly with the learning ratios, while the GV/BVPRE was negatively related to the learning rates. GVPRE and DFAPRE did not correlate with learning ratio in any retention test.

In order to fully understand the relationship between motor variability and learning rate avoiding initial performance bias, as it has been mentioned in the statistical analysis section, participants were grouped according to the participants’ GV/BV using a linear regression method [1]. DFA was not used for this analysis as it did not show any significant relationship with learning rate nor initial performance. The participants with higher residual scores in each performance level were included in the Higher-GV/BV group, whereas the participants with lower residual scores were included in the Lower-GV/BV group. Some differences in the effect of practice were found between GV/BV groups (Figure 5). Regarding the participants’ performance, only the Higher-GV/BV group displayed higher HR in all the tests after training compared with the Pretest (Figure 5A). Concerning GV values, only the Higher-GV/BV group showed significant differences, decreasing its values in the Posttest, Retest1, and Retest2 compared with the Pretest. However, that reduction was not maintained after one week of resting, being significantly different in Retest3 compared with the Posttest and Retest1 (Figure 5C). Both groups significantly decreased their BV values after training. The Lower-GV/BV group displayed significantly lower values in the Posttest (*p* = 0.009) and Retest2 (*p* = 0.006) compared with the Pretest, while the Higher-GV/BV group showed these differences in the Posttest (*p* = 0.004) and Retest1 (*p* = 0.025), returning to the initial values in Retest2 (*p* = 0.014) and Retest3 (*p* = 0.002) as compared to Retest1 (Figure 5D). No effect of training was found in GV/BV for any group (Figure 5B).

## 4. Discussion

Motor variability plays a functional role in human adaptive behaviors, being related to the facilitation of motor learning [1,2]. However, this relationship is not fully understood because variability regulation during the learning process depends on both the individuals’ and the environment’s characteristics [6,7]. The motor synergies variability has been related to the individual’s learning stages in which different strategies prevail (i.e., exploration and/or exploitation); however, it has not been directly related to the learning rate. In addition, a previous study indicated that motor variability structure predicted the learning rate in error-based learning tasks [1], but to the best of authors’ knowledge, there is no such evidence in reward-based learning tasks. In this study, a computer simulation task was designed and a force dynamometer was used to measure movement variability during the performance of a virtual throwing task. Thus, the main aim was to assess the relationship between motor variability and learning rate in a reward-based learning task analyzing the features of motor variability through its structure and motor synergies characteristics.

First of all, we observed a learning effect in the reward-based task carried out in the study. A significant improvement in performance (i.e., higher hit ratio values) was found after training and it remained stable along all the retests (Table 1 and Figure 4). This improvement was accompanied by an initial decrease in the force applied to release the ball and a reduction in good and bad variability, while the variability of the motor synergies remained stable. Regarding the GV/BV, the findings were according to one of the learning stage scenarios in which both good and bad variability decreased in a proportion such that the relative difference between them did not change [21]. These results have also been found in other experimental studies in which two-arm pointing tasks based on error-based learning were assessed [30]. These changes returned to the initial levels after the resting period while maintaining the performance improvement. In the aforementioned study, kinematic data for both arms were collected, providing direct information about how the DoFs related to motor coordination were evolving. This is a limitation in our study. Our results are based only on the performance output, no kinematic information was included. On the other hand, in the study by Domkin et al. (2005) there were no retests to check the variability evolution of the motor synergies in retention. Future studies should address the whole evolution of the learning process, providing kinematic information to check the DoFs’ evolution.

Subsequently, we carried out correlational analyses to assess if the variability in motor synergies or the variability structure of error time series could predict the learning rate.

First of all, we found that those participants showing a higher variability of the motor synergies also displayed a lower learning rate. In addition, participants with higher initial bad variability levels obtained higher learning rates. Based on these results, higher bad variability levels could be interpreted as an index of exploratory behavior in which the participant would be looking for the right motor solutions. Thus, variability magnitude parameters such as those obtained from the goal equivalent manifold approach, would reflect the learners’ capacity to explore different motor configurations (i.e., synergies) until they find the reward zones in a motor task [30,31]; that is, to find the optimal motor solution to achieve the task goal [2]. Conversely, these results can also be interpreted as the high levels of bad motor variability impair motor learning and should be reduced. Based on the experimental study carried out by Cardis et al. (2018) [32], increasing motor variability may adversely affect the ability to retain the learned solution.

Analyzing the other variability tool used in this study, even though DFA has proven to be useful for predicting the learning rate in the error-based continuous task [1], it did not correlate to the learning ratio in the reward-based task of this study. The results showed that the DFA values of the fluctuations did not predict the learning rate in this reward-based task. Previous studies have supported the use of the long-range autocorrelation index to elicit the relevant role of the motor variability during motor error-based learning, that is, how learners dynamically adjust their movement according to the task demands. Nevertheless, this temporal dynamic of the outcome variations revealed by DFA does not seem to be a determinant index of the learners’ success to aim at the target in our reward-based task. This could be explained by the fact that the success in error- and reward-based tasks depends on different learning strategies that can be revealed through different tools. The learning processes of an error-based task would mainly depend on the error sensitivity, that is, the ability to detect variations between the desired behavior and the actual motor outcome [33]. The relationship between the learning rate and motor variability measured by the ratio between the good and bad variability instead of the DFA observed in this study would support the idea that the modulation of the magnitude of motor variability (i.e., exploring different motor solutions) is more relevant to foster learning processes in reward-based learning tasks than enhancing error sensitivity. In spite of this rationale, it must be pointed out that the lack of prediction capability shown by the DFA in this study could also be related to the fact that this tool usually needs longer trial-to-trial variation datasets to provide a reliable score [34,35,36].

Focusing on the goal equivalent manifold approach parameters, it must be taken into consideration that the correlation between the variability of the motor synergies and learning rate should be taken with caution because motor synergies variability was highly related to the initial performance level too. Thus, the participant´s initial performance level biased the relationship between motor synergies variability and learning ratio. This is, people with higher bad variability or lower motor synergies variability would show higher learning because they also showed poorer initial performance, and, thus, they have larger room for improvement. A similar bias was also found by Barbado et al. (2017) using DFA to predict the learning rate in an error-based learning task. Following Barbado et al.’s suggestions for reducing the bias caused by the initial performance, participants were grouped based on the linear regression between their HR in the pretest and their learning rate. Contrary to the results observed in the correlational analyses, those participants with initial higher motor synergies variability displayed a greater performance improvement than those with initial lower motor synergies variability. When we reduced the effect of initial performance effect in learning rate, we found that the participants who showed larger good variability compared to bad variability in the pretest (i.e., Higher-GV/BV group) also showed higher learning rate. The interpretation of these findings could be that participants from the Higher-GV/BV group are exploring among the range of successful motor synergy combinations to achieve the task goal. Therefore, it can be concluded that the larger spectrum of motor configurations displayed for these participants would help them to find the reward zone easily. It must be pointed out that the evolution of the motor synergies variability was similar in both groups, although to a different extent. The results found in the two groups supports those from the whole cohort. Both groups displayed no changes in their motor synergies variability, because both good and bad variability decreased in a proportion such that the relative difference between them did not change [21]. Participants from the group with lower variability of the motor synergies decreased their bad variability after training but no significant changes were found in their good variability levels. However, participants with higher initial levels of motor synergies variability decreased both bad and good variability. In both cases, although the changes were not significant, an increase in the motor synergies variability could be appreciated, which could be interpreted as a step forward to strengthen and stabilize the synergies found by the participants to achieve the task goal [21]. Finally, after the resting period both good and bad variability moved closer to the initial values. To the best of our knowledge, no previous studies have addressed the evolution of the motor variability synergies after a resting period without practice. According to the aforementioned learning stages, Latash (2010) indicated that this increment of motor variability in the retention tests could be related to the first stage, which could mean that participants are unlearning; however, the performance displayed by the participants did not decrease in those retention tests. Then, our interpretation, being aware that there is no information about the kinematic data, is that participants went from the stabilization stage achieved right after practice to a flexibilization of their synergies.

## 5. Limitations

It is important to point out that extrapolation of the results of this study is limited. First of all, the task studied in this work is a simple task performed in a computational environment. Second, as mentioned in the manuscript, how the individual manipulates motor variability to promote a faster learning rate depends on the task and the individual features. Thus, in order to extrapolate the results, other studies using other kind of reward-based learning tasks in a more realistic environment are needed.

## 6. Conclusions

Our findings showed that analysis of the motor variability synergies using the procedure applied could reveal the relevant role of motor variability during motor reward-based learning. The evolution of the motor variability synergies proposed by different authors and related to different learning stages seems to be supported by our study, which also provides information about the retention periods. Future studies should assess if this way to characterize motor variability, as some other tools did, such as DFA, could also be useful to predict motor error-based learning.

## Figures and Tables

**Figure 1 sensors-21-06448-f001:**
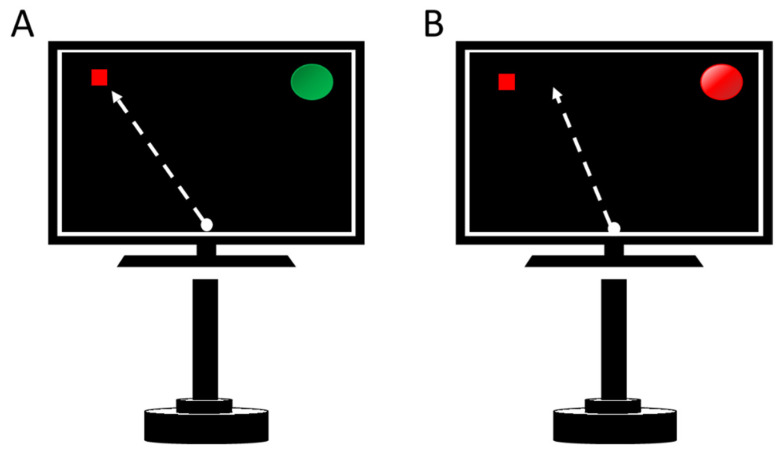
(**A**) Visual information when the participant hit the target; (**B**) Visual information when the participant did not hit the target. The ball trajectory was occluded during the experiment; the trajectory is shown in the figure as a dotted line only for the description of the task.

**Figure 2 sensors-21-06448-f002:**
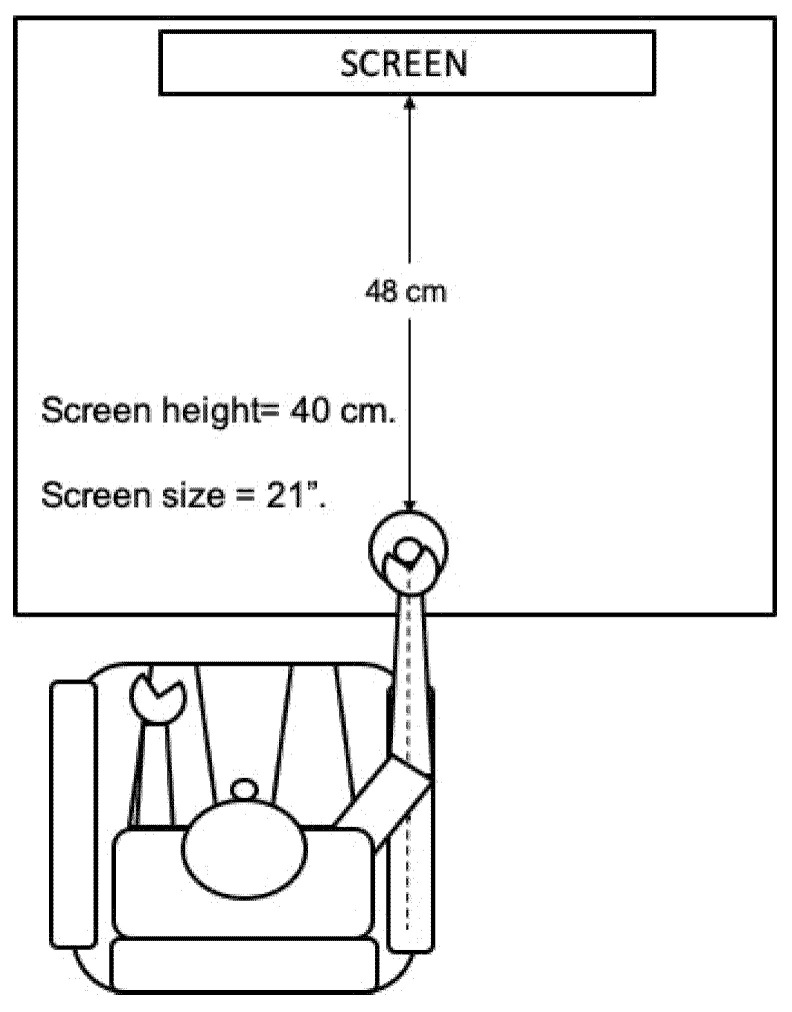
Participant position and instrument distribution during the performance of the task.

**Figure 3 sensors-21-06448-f003:**
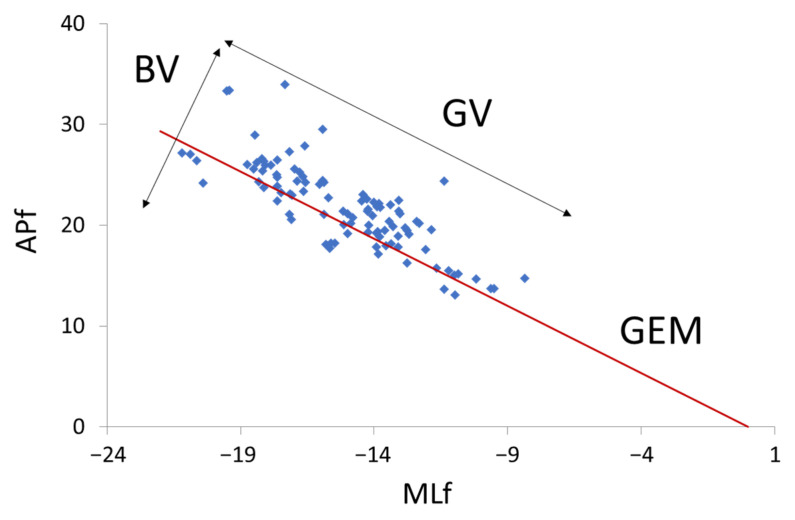
Representation of the different motor synergies variability. GV: Good variability; BV: Bad variability; MLf: force in medial–lateral axis; APf: force in anterior–posterior axis.

**Figure 4 sensors-21-06448-f004:**
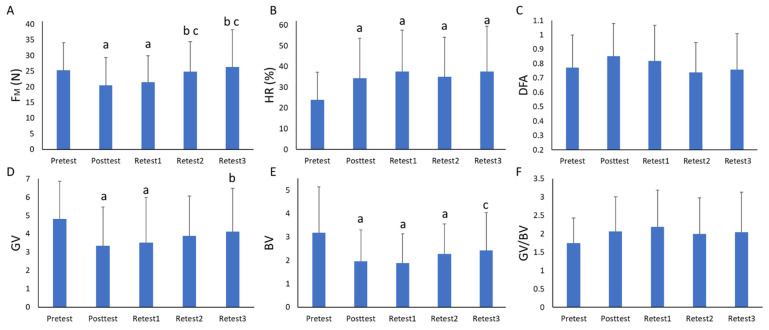
Average error values in all the tests calculated in the study and the pair comparisons results extracted from the repeated measures ANOVA. (**A**) refers to the Force magnitude values; (**B**) refers to the Hit ratio values; (**C**) refers to Detrended Fluctuation values; (**D**) refers to Good Variability values; (**E**) refers to Bad Variability values; and (**F**) refers to the Good and Bad variability ratio. a = significant differences compared with the pretest; b = significant differences compared with the posttest; c = significant differences compared with retest1.

**Figure 5 sensors-21-06448-f005:**
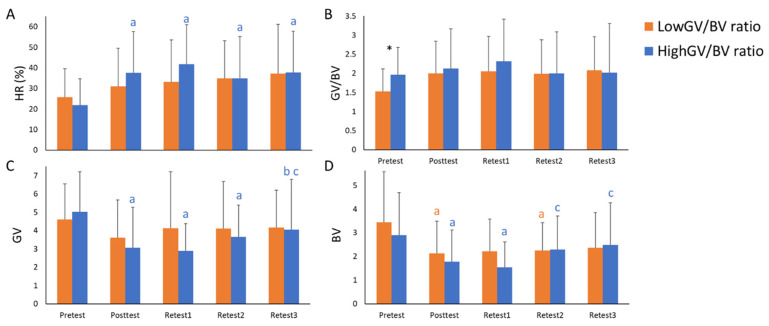
Pair comparisons results between the groups with different initial levels of GV/BV. (**A**) the Hit ratio values; (**B**) refers to the Good and Bad variability ratio; (**C**) refers to Good Variability values; and (**D**) refers to Bad Variability values. * = significant differences between groups; a = significant differences compared with the pretest; b = significant differences compared with the posttest; c = significant differences compared with Retest1.

**Table 1 sensors-21-06448-t001:** Average error values (mean ± SD) in all the tests calculated in the study and repeated measures ANOVA statistics for the effect of training in the experimental variables.

Variables	Pretest	Posttest	Retest 1	Retest 2	Retest 3	F	*p*	η^2^
FM (N)	25.32 ± 8.76	20.42 ± 8.98	21.43 ± 8.54	24.83 ± 9.61	26.37 ± 11.84	10.859	<0.01	0.202
HR (%)	23.88 ± 13.31	34.35 ± 19.33	37.50 ± 20.01	34.92 ± 19.13	37.50 ± 21.83	10.644	<0.01	0.198
GV	4.81 ± 2.06	3.34 ± 2.12	3.51 ± 2.46	3.88 ± 2.17	4.11 ± 2.38	8.062	<0.01	0.158
BV	3.18 ± 1.97	1.95 ± 1.35	1.88 ± 1.25	2.27 ± 1.29	2.43 ± 1.61	12.224	<0.01	0.221
GV/BV	1.74 ± 0.69	2.06 ± 0.94	2.19 ± 1.01	2.00 ± 0.98	2.05 ± 1.09	2.289	0.062	0.051
DFA	0.77 ± 0.23	0.85 ± 0.23	0.82 ± 0.25	0.74 ± 0.21	0.76 ± 0.25	2.044	0.090	0.045

FM = Force magnitude; HR = Hit ratio; GV = Good variability; BV = Bad variability; GV/BV = the ratio between good variability and bad variability; DFA = DFA alpha value of the relative error time series data.

**Table 2 sensors-21-06448-t002:** Pearson product moment correlation coefficient calculated between initial performance (HRPRE), the learning ratio, and the initial outcomes of the rest of the variables.

	HR_PRE_ (%)	LR_POST_	LR_RET1_	LR_RET2_	LR_RET3_
HR_PRE_ (%)	--	−0.504 **	−0.579 **	−0.417 **	−0.558 **
FM_PRE_ (N)	−0.201	−0.019	0.023	0.018	0.094
GV_PRE_	−0.340 *	0.140	0.180	0.080	0.195
BV_PRE_	−0.618 **	0.320 *	0.370 *	0.254	0.477 **
GV/BV_PRE_	0.667 **	−0.277	−0.341 *	−0.263	−0.392 **
DFA_PRE_	0.207	−0.068	−0.067	−0.238	−0.138

* *p* < 0.05; ** *p* < 0.01; HR_PRE_ = Hit ratio in the pretest; FM_PRE_ = Force magnitude in the pretest; GV_PRE_ = Good variability in the pretest; BV_PRE_ = Bad variability in the pretest; GV/BV_PRE_ = Good variability/Bad variability ratio in the pretest; DFA_PRE_ = DFA from the relative error in the pretest. LR_POST_ = the differences in the HR between the pretest and the posttest; LR_RET1_ = the differences in the HR between the pretest and retest1; LR_RET2_ = the differences in the HR between the pretest and retest2; LR_RET3_ = the differences in the HR between the pretest and retest3.

## Data Availability

The data presented in this study are available in the Appendix A here.

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
