# Peer review of "Motor Synergies Measurement Reveals the Relevant Role of Variability in Reward-Based Learning"

_sensors, 2021, doi:10.3390/s21196448_

Round 1
Reviewer 1 Report
General Comments
The authors investigated how force variability is structured during learning of a computer-based ball throwing task. In general, this was an interesting study that addresses an important question related to the relationships between variability and task performance. I enjoyed reading this manuscript. However, I do have some concerns that should be addressed, mostly related to the theoretical underpinnings and motivation of the study, as well as some of the analysis methods. See below for major and minor comments.
Major Comments
-Introduction: The introduction could benefit from a clearer overview of the current literature. There are many different concepts discussed, such as low/high variability, error- and reward-based tasks, non-linear analysis methods, etc. but overall the introduction is pretty disjointed. In its current state, I think the readership would be confused as to what the current state of the literature is and where the gap/need for study is. As such, a rewrite/reorganization of the introduction would be greatly beneficial to this section of the manuscript.
-Introduction: What were the hypotheses the authors pursued in this study? Including these in the introduction would allow for readers to more clearly understand why the experiment was designed the way in which it was.
-Introduction, Lines 46-64: Please elaborate more on error-based and reward-based tasks. The authors describe how error-based tasks should be avoided in motor learning studies such as the current one, but then describe tasks with accuracy constraints such as shooting or reaching. These are generally controlled through feedforward mechanisms and therefore utilize trial-and-error to correct movement patterns. Perhaps a clearer definition of error-based and reward-based tasks would be helpful here, including greater explanation as to why error-based tasks should be avoided.
-Methods: The description of computation of good variability vs. bad variability is well done, however more elaboration may be warranted. In its current state, it is somewhat unclear how these variability parameters were calculated. For example, how was the task variable (GEM) determined based on accuracy?
-Methods, Statistical Analysis: Why were participants grouped according to their initial performance, if they were all novices to begin with? Simply citing a method in a different article does not quite justify this decision, which has major implications for the remainder of the study. If the sample is generalizable to a broad population, at least a greater description as to why this analysis decision was made is warranted. It would be interesting to see how this analysis changes if the subgrouping does not occur.
-Discussion: There is no mention of the limitations to this study, which is important to discuss especially in the context of how generalizable these findings are to more realistic tasks. Is the noted variability structure task-specific, or can it apply to other types of tasks?
Minor Comments
-Introduction, Lines 30-32: This sentence is unclear, please revise
-Introduction, Line 39: Please revise this sentence to be past tense, otherwise it may be confusing to the readers as it reads like this study has not yet taken place
-Introduction, Lines 73-77: How specifically does the GV/BV ratio predict stage of learning or exploration/exploitation of motor learning strategies? Perhaps more detail is warranted here.
-Methods, Line 99: Should ‘sensibility’ read as ‘sensitivity’, in this instance?
-Methods: Why was force magnitude computed, and how did it relate to task accuracy?
-Results, Table 1: It would be interesting to see the effect sizes (e.g., partial eta^2) for each statistical comparison made. This would further explain some of the significant statistical findings.
-Figures 4 and 5: Consider labeling each subplot for easier reference in the figure caption (e.g., Figure 4a, 4b, etc.). Also, for the DFA plot (Figure 4) is It possible for the value to be > 1, as indicated on the y-axis?
-Discussion: Are there any factors related to participants (e.g., prior video game use) which made this task easier?
Author Response
Please, see the attached file. Thank you so much for your review.

Reviewer 2 Report
The paper investigates whether different variability measures correlate with learning rate in a simple virtual-throwing task. It is always good to try to understand how variability relates to improvements in practice as the theme has not been fully understood by the literature. The paper does its job in relating the variables despite there is a need for a deeper discussion on the measures and possible inferences.
I have three major concerns with the paper:
1) How can the paper move beyond simplistic explanations on variability/exploration and relate to learning? That is, if BV is positively related to learning rate, then, it is about exploring different solutions; if GV is related to learning rate, then, it is about exploring within the same solution. There is no alternative on which exploration is not necessary or variability is not good. To discuss possibilities, I would invite the authors to read some interesting papers from Rajiv Ranganathan that shows that inducing more variability (in null or goal spaces) does not lead to better results (Cardis et al., 2018 - High variability impairs motor learning regardless of whether it affects task performance); or that those who vary less are the most adaptive (Ranganathan & Newell, 2010 - Emergent flexibility in motor learning).
2) I think the task requires some justification and elaboration. It seems overly simple and might be that the direction of push was sufficient for good performance. The fact that GV/BV does not change over time seems, at least to me, that the task was so simple that individuals passed "stage 1" from Latash (2010) ideas where individuals need to learn the structure of the task. I know that similar experiments are done within computational approaches to motor behavior, but the idea there is that the individual is performing something "well-learned" already and adaptations are solely based on outcomes (error-based, rewards, etc.).
It might be that improving the task description might help as well. How the push relates to the task? the handle was fixed? Paritcipants maintained holding the handle or just pushed it when the task required? I am missing the step-by-step of the trials and the exact relationg between "pushes" and task variables.
3) I understand that the methods seem to follow what was performed elsewhere, but the paper should stand on itself. I did not understand the regression and residuals analyses that gave rise to the main conclusions of the study. The authors should make an effort to improve description on it.
Minor points:
- The authors tend to use the argument of "it has not been done yet" several times through the text. One should be cautious to use that as the main argument for the paper.
- Lines 77-79: The authors argue that there was not a paper relating variability with reward-learning. But note, there are papers on motor learning in general with UCM (including ones cited in Latash, 2010). These papers can be compared and discussed despite they did not differentiate reward and error-based learning.
Thus, the authors must cite these papers and discuss how the current paper differentiates from them and advances knowledge in the field. That is, how reward learning could be different? How measures of variability relate to the task in this case that is different from some other paradigms? - Line 100: I did not understand what is meand by "it provides a small deflection in the sensor system".
- I think that the paper has too many acronyms. The authors should choose better. Force magnitude does not need to be shortened, neither hit ratio. GV/BV can be called synergy index without shortening as well.
- Why should learning ratio be considered in terms of post test, ret 1, ret 2, and ret 3? The authors should have a rationale and follow it. For instance, if the authors want to relate to exploration, then LR postest is sufficient (because it would be about improving performance given exploratory behavior).
- Lines 284-310: The authors should try to organize the paragraphs according to themes. For now, I see too many variables and their theoretical points being discussed in a single paragraph.
- Lines 292-300: I don't understand the DFA explanation. How "assessing the temporal dynamic of the outcome variations as an index of how the learners modified their behavior" is not important in the present task?
- Lines 342-347: The authors speculate and do not explain what is meant by their statements in this part: "From the interpretation of the increment of motor variability in the retention tests it could be understood that participants are unlearning; however, the performance displayed by the participants did not decrease in those retention tests. Then, our interpretation is that participants went from the stabilization stage achieved right after practice to a flexibilization of their synergies." This should be properly unpacked.
I hope this helps.
Matheus Pacheco
Author Response

(The authors gave the same response as above.)

Round 2
Reviewer 2 Report
I believe the authors responded to my concerns. I hope to have helped them improve the quality of the paper.